# Free Appropriate Public Education, the U.S. Supreme Court, and Developing and Implementing Individualized Education Programs

**Michael Rozalski [1,\*], Mitchell L. Yell [2] and Jacob Warner [3]**

1   Ella Cline Shear School of Education, State University of New York at Geneseo, South Hall 219C, Geneseo, NY 14454, USA
2   Wardlaw College of Education, University of South Carolina, Columbia, SC 29208, USA; MYELL@sc.edu
3   Broome-Tioga Board of Cooperative Educational Services, Binghamton, NY 13905, USA; warnerj37@gmail.com
\*   Correspondence: rozalski@geneseo.edu

**Abstract:** In 1975, the Education for All Handicapped Children Act (renamed the Individuals with Disabilities Education Act in 1990) established the essential obligation of special education law, which is to develop a student's individualized special education program that enables them to receive a free appropriate public education (FAPE). FAPE was defined in the federal law as special education and related services that: (a) are provided at public expense, (b) meet the standards of the state education agency, (c) include preschool, elementary, or secondary education, and (d) are provided in conformity with a student's individualized education program (IEP). Thus, the IEP is the blueprint of an individual student's FAPE. The importance of FAPE has been shown in the number of disputes that have arisen over the issue. In fact 85% to 90% of all special education litigation involves disagreements over the FAPE that students receive. FAPE issues boil down to the process and content of a student's IEP. In this article, we differentiate procedural (process) and substantive (content) violations and provide specific guidance on how to avoid both process and content errors when drafting and implementing students' IEPs.

**Keywords:** individualized educational program; law/legal issues; policy; procedural; substantive

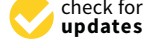



## 1. Free Appropriate Public Education, the U.S. Supreme Court, and Developing and Implementing Individualized Education Programs

Enacted in 1975, the Education of All Handicapped Children Act (EAHCA) set the federal standard for the education of eligible students with disabilities in the United States. Since 1975, the EAHCA has gone through a number of reauthorizations and amendments, including in 1997 when it was retitled the Individuals with Disabilities Education Act (IDEA), and again in 2004. Although these amendments somewhat altered sections of the law and fine-tuned the procedures for special education, the fundamentals have remained the same.

The EAHCA was established to provide federal money to state and local education agencies to educate eligible children and youth with disabilities in accordance with the federal law. For a state to qualify for assistance, the state had to demonstrate that it had a policy in effect that ensured that all eligible students with disabilities would receive a free appropriate public education (FAPE) in accordance with the terms of the law (Yell 2019). Thus, a federal commitment to students with disabilities was established both in terms of the educational rights of students with disabilities and as a fiscal partnership with the states (Martin 2013; Stafford 1978). Congress established the individualized education program (IEP) as the vehicle for actualizing an individual student's FAPE. The purpose of this article is to (a) examine the FAPE mandate of the IDEA, focusing on the procedural, substantive; and implementation requirements; (b) differentiate procedural and substantive violations;

and (c) offer guidance on crafting IEPs that avoid these errors, meet the FAPE requirements of the IDEA, and provide academic and functional benefits to students with disabilities.

## 2. The FAPE Requirements of the IDEA

When the EAHCA was passed and signed by President Gerald Ford in 1972, the law required that all eligible students with disabilities be provided with a FAPE. The definition of FAPE in the EAHCA was special education and related services that would (a) be provided at public expense, under public supervision and direction, and without charge, (b) meet standards of the State educational agency, (c) include an appropriate preschool, elementary, or secondary school education in the state involved, and (d) be provided in conformity with the individualized education program (IDEA, 20 U.S.C. § 1401 (a)(18)). This definition has remained unchanged since the original passage of the law. According to a Congressional author of the EAHCA, "We in Congress did not attempt to define 'appropriate' in the law but instead, we established a baseline mechanism, a written document called the individualized education program (IEP)" (Stafford 1978, p. 75), This document, the IEP, therefore, provides the foundation and the blueprint for measuring and evaluating the education provided to students with a disability. As the baseline mechanism that ensures a FAPE for eligible students, each IEP must be designed to meet the unique individual needs of each student. School professionals and a student's parents must be involved in crafting the student's IEP. According to the U.S Supreme Court, the IEP is the "modus operandi of the law" (*Burlington School Committee v Massachusetts Department of Education*, p. 363).

Despite the specification that school personnel and a student's parents collaborate in the development of the student's IEP, whether the IEP actually confers a FAPE has sometimes been a controversial issue. As the Congressional authors of the IDEA recognized disputes would occur regarding a student's special education program in his or her IEP, they included resolution mechanisms whereby parents, and sometimes school district officials, could request a review by an impartial due process hearing officer. Following a due process hearing, either party could appeal part or all of a decision to state or federal court. As neither the language of the EAHCA nor the language of the regulations implementing the law specified what constituted an appropriate education, it would be up to the courts, and especially the U.S. Supreme Court, to clarify how IEP teams could ensure the special education programs they develop and implement provide students with a FAPE (Conroy and Yell 2019; Crockett and Yell 2008). In fact, the U.S. Supreme Court has heard two cases interpreting the FAPE mandate of the IDEA: *Board of Education v. Rowley* (1982; hereinafter *Rowley*) and *Endrew F. v. Douglas County School District* (2017; hereinafter *Endrew* F.). We next briefly review these cases and what they mean for school district personnel having to develop and implement students' IEPs.

## 3. The U.S. Supreme Court and FAPE

### 3.1. Board of Education v. Rowley (1982)

Amy Rowley was a young girl who was deaf. She attended the Furnace Woods Elementary School in the Hendrick Hudson School District in Peekskill, NY. Prior to attending kindergarten, Amy's parents and school officials met and decided to place Amy in a regular kindergarten class to determine what services she would need. School personnel took measures to prepare for Amy's arrival at the public elementary school, such as having school officials take courses in sign language interpretation and providing devices to facilitate communication with Amy's parents who were also deaf. Amy was also given an FM hearing aid, which amplified words spoken into a wireless receiver by the teacher or fellow students during certain classroom activities.

An IEP was prepared for Amy during the fall of her first-grade year. She would remain in a regular classroom during her first-grade year and would continue to use the FM hearing aid. She would also receive instruction from a tutor for the deaf for one hour each day and from a speech therapist for three hours each week. The Rowleys agreed

with parts of the IEP but insisted that Amy also be provided a qualified sign language interpreter in academic classes. School district officials agreed to provide an interpreter in Amy's kindergarten class for an experimental period, but after one week, the interpreter had reported that Amy did not need his services at that time. After consulting with the school's Committee on the Handicapped and collecting testimony from teachers and other school personnel, the school's administrators decided that Amy did not need the interpreter while in first grade. After the request for the personal interpreter was denied, the parents requested a hearing from an examiner who sided with the district, and agreed Amy was indeed receiving a FAPE.

Rowley was eventually heard before the U.S. district court. The judge in the case noted that Amy had an IQ of 122, performed above average academically in class, was well-adjusted, had many friends among her peers, and interacted well with her teachers. The judge also determined the FAPE mandate of the EAHCA required "each handicapped child be given an opportunity to achieve his full potential commensurate with the opportunity provided to other children" (*Rowley*, 1980, p. 534). The court ruled even though evidence established Amy was receiving an adequate education, district officials failed to consider "the importance of comparing her performance to that of non-handicapped students of similar intellectual caliber and comparable energy and initiative" (*Rowley*, 1980, p. 534). Thus, the court ruled that the by not providing Amy with a sign language interpreter, the school district had failed to provide Amy with a FAPE. The school district appealed to the Second Circuit, which affirmed the lower court's ruling. Officials in the Henrick Hudson School District then exercised their final appeal to the U.S. Supreme Court. The appeal was granted and on March 23, 1982 oral arguments were made in the first special education case to be argued before the High Court. In this case, the Supreme Court considered two primary questions, (a). What was meant by the law's FAPE requirement, and (b) what is the role of state and federal courts in exercising a review under the EAHCA.

During the case, a primary issue that the Justices struggled with was the absence of any clear definition of what an "appropriate" education is for students with a disability. They concluded the main intent of the Act was "more to open the door of public education by the means of specialized education services than to guarantee any substantive level of education once inside" (*Rowley*, 1982, p. 177). Chief Justice William Rehnquist noted in his majority opinion that the law "set forth extensive procedures to be followed in formulating personalized education programs" (*Rowley*, 1982, p. 192) and "that adequate compliance with the procedures prescribed would in most cases assure much if not all of what Congress wished in the way of substantive content in an IEP" (*Rowley*, 1982, p. 204). As a result, schools were required to provide access to education for students with a disability but were not required to ensure these students were achieving at the same level as their peers (Wenkart 2009).

The High Court established a two-step test for determining if a school district had met the FAPE standard of the EAHCA. The test, which would be used by independent hearing officers (IHOs), administrative law judges (ALJs), and state and federal judges when ruling in FAPE cases, required that the first questions asked was if the school district complied with the procedures set forth in the Act and second, was if the individualized educational program developed through the Act's procedures was reasonably calculated to enable the child to receive educational benefits? The first part of the *Rowley* test was procedural and stressed the importance of schools adhering to the procedural requirements of the IDEA when determining if a school had provided a FAPE. The second part of the *Rowley* test was substantive and required courts to examine a student's IEP to determine whether the IEP developed by the school was reasonably calculated to enable a student to receive educational benefits. The High Court recognized the more difficult nature of this second part of the test: "The determination of when handicapped children are receiving sufficient educational benefits to satisfy the requirements of the Act presents a more difficult problem" (*Rowley*, 1982, p. 198). Nonetheless, the according to the U.S. Supreme Court, "If these requirements are met, the State has complied with the obligations imposed by Congress

and the courts can require no more" (*Rowley*, 1982, p. 207). The Supreme Court concluded the school district had indeed met the requirements of the two-part FAPE test to Amy and had indeed complied with the language of the EAHCA.

As the Supreme Court did not "establish any one test for determining the adequacy of educational benefits conferred upon all children covered by the Act" (*Rowley*, 1982, p. 204), the lower courts began to adopt different standards to determine what degree of educational benefit was necessary to provide FAPE. According to Yell and Bateman (2017), some courts of circuit courts adopted a higher standard by which to determine the educational benefit needed to confer a FAPE. However, most circuit courts adopted a lower standard of educational benefit to determine a FAPE. Courts that used a lower standard often ruled that if a school district provided some degree of educational benefit, no matter how small, the district conferred a FAPE (Johnson 2012; Seligmann 2017; Yell and Bateman 2017). This division among the circuit courts made it likely that eventually the High Court would hear another FAPE case. That opportunity presented itself 35 years after the Supreme Court's ruling in *Rowley*. The FAPE case appealed to the Supreme Court was out of the U.S. Court of Appeals for the Tenth Circuit: *Endrew F. v. Douglas County Public School District* (2015).

*3.2. Endrew F. v. Douglas County School District* (2017)

Anyone who has spent any length of time in the educational world knows that there is never a set standard by which all students or schools can be measured. Special education and the IDEA itself are built on the notion that students benefit from individualized and specialized programs that uniquely fit a student's needs (Aron 2005). The standard set by Rowley is still an important measuring stick which an IEP must stand up to, but the recent case of *Endrew F.* has altered that standard (Davidson 2016; Waterstone 2017; Zirkel 2019).

Endrew F., who was called Drew by his parents, was diagnosed with autism and attention-deficit hyperactivity disorder (ADHD) disability from a young age. Drew had attended the Summit View Elementary School in the Douglas County School District from pre-k through fourth grade and had been furnished with an IEP every year. However, by Drew's fourth grade year, his parents were dissatisfied with the progress he was making and believed that his IEP was too similar to his previous IEPs. As a result, the parents withdrew him from the district and enrolled him in a private school, the Firefly Autism House. While at the private school, a behavioral intervention plan was developed which addressed many of Drew's challenging behaviors and he began to make academic gains that he had not made in the public school setting. As Drew's parents wanted him to attend the Douglas County School District, they and the school personnel developed a new IEP for Drew. Unfortunately, Drew's parents decided that the new IEP was too similar to his previous fourth grade IEP and did not include behavioral interventions. His parents then filed a complaint seeking tuition reimbursement from the school to compensate for Drew's education at the private school. In order for the parents to receive this reimbursement, they would need to show that the school district had not provided Drew with a FAPE.

As Drew's case worked its way up through the due process hearing, district court and U.S. Court of Appeals for the Tenth Circuit, the low educational benefit standard set by the tenth circuit court was applied at each level. According to the tenth circuit's education benefit standard, the school district had provided educational benefit that was more than de minimis, thus providing him with a FAPE. Following the ruling against them in the circuit court, Drew's parents filed an appeal with the U.S Supreme Court. The main question pertaining to this case was what is the level of educational benefit that school districts had confer on students with disabilities to provide them with a FAPE as guaranteed by the IDEA? The High Court accepted the appeal and heard the case on 17 January 2017.

Chief Justice of the Supreme Court John Roberts wrote the unanimous ruling for the Court, which was issued on 22 March 2017. In the opinion, Justice Roberts wrote that in *Rowley* the High Court had declined to endorse any one standard to assist lower courts to determine when students with disabilities are receiving sufficient educational benefits to

satisfy the FAPE requirements of the IDEA but noted "that more difficult problem is before us today" (*Endrew F.*, 2017, p. 993). The new educational benefit standard announced in the Endrew F. ruling was "to meet its substantive obligation under the IDEA, a school must offer an IEP reasonably calculated to enable a child to make progress appropriate in light of the child's circumstances" (*Endrew F.*, 2017, p. 1001). Justice Roberts asserted that the "the essential function of an IEP is to set out a plan for pursuing academic and functional advancement" (*Endrew F.*, 2017, p 992). He further wrote that the new educational benefit standard was "markedly more demanding than the merely more than *de minimis*" test applied by the Tenth Circuit . . . " (*Endrew F.*, 2017, p. 992). In its ruling, the Supreme Court vacated or annulled the tenth circuit court's previous ruling in *Endrew F.* and instructed the tenth circuit court to reissue its ruling in light of the new standard.

The tenth circuit court then remanded the ruling in *Endrew F.* to the district court. According to the district judge, Lewis Babcock "the April 2010 IEP offered to (Drew) by the District was insufficient to create an educational plan that was reasonably calculated to enable Petitioner to make progress, even in light of his unique circumstances, based on the continued pattern of unambitious goals and objectives of his prior IEPs" (*Endrew F.*, 2018, p.16). Furthermore, the judge noted that Drew's IEP "was clearly just a continuation of the District's educational plan that had previously only resulted in minimal academic and functional progress" (*Endrew F.*, 2018, p.16). Furthermore, the judge noted that the district's failure to address Drew's behaviors "impacted his ability to make progress on his educational and functional goals" (*Endrew F.*, 2018, p.17). Thus, district court judge overturned his ruling and held that under the Supreme Court's new educational benefit standard, the Douglas County School District had failed to provide Drew with a FAPE and the Firefly Autism House had provided a good education. Drew's parents, therefore, were intitled to reimbursement tuition from the district to cover the costs of the private school and attorneys' fees (Turnbull et al. 2018). The award for the costs incurred at the private facility and reasonable attorneys' fees and litigation costs, amounted to $1.3 million dollars (Aguilar 2018).

## 4. Procedural, and Substantive Violations of FAPE

The two seminal Supreme Court decisions on FAPE, *Rowley* and *Endrew F.*, set forth a two-part test that impartial hearing officers (IHOs), Administrative Law Judge (ALJs), and state and federal judges must apply to the facts of the case when determining if a school district provided a FAPE. The first part of the test, the procedural litmus test, was from the *Rowley* ruling. In this part of the test, the IHO, ALJ, or judge determines if the school district followed the procedures of the IDEA. The IDEA includes a number of procedural requirements to which school district personnel must adhere (e.g., ten-day notification required for parents prior to a meeting). Additional guidance has been provided by cases heard in the 3rd (e.g., *Ridley School District v. M.R. and J.R. ex rel. E.R.,* 2012; *D.B. v. Gloucester Township School District*, 2012), and the 9th Circuits (e.g., *M.L. v. Federal Way School District*, 2004). Although in many cases these requirements are straightforward and easily understood, school districts still make common procedural errors, examples of which can be found in Figure 1.

The second part of the test, which is the substantive or educational benefit component, is from the *Endrew F.* ruling. Recall in the Rowley test the second part of the test was a follows: Was a student's IEP reasonably calculated to enable the student to receive educational benefit? The Endrew standard changed the second question as follows: Was a student's IEP reasonably calculated to enable a student to make progress appropriate in light of his or her circumstances? Thus, in the second part of the test, the HO, ALJ, or judge determines if students' IEP meets this new educational benefit standard and if the IEP confers more than trivial or de minimis benefit to a student.

---

**Common Procedural Errors**

- Failing to provide prior written notice

- Failing to ensure parents' meaningful involvement

- Predetermining services and placement

- Improper IEP membership

- Failing to ensure a continuum of alternative placements

- Determining placement prior to making programming decisions

- Failing to address transition needs and services

- Failing to implement the IEP as written

---

**Figure 1.** Procedural Errors.

Recall also that the Supreme Court in both *Rowley* and *Endrew F.* addressed the difficulty of determining the degree of educational benefit required in order to meet the FAPE requirements of the IDEA. This is largely because the IDEA provides no specific information on the substantive requirements. Nonetheless, the courts, particularly the Supreme Court in *Rowley*, have divided the FAPE requirement into two components, the procedural requirements and substantive requirements. Moreover, in the enactment of the Individuals with Disabilities Education Improvement Act in 2004, Congress emphasized the procedural and substantive distinction in matters involving a violation of a student's FAPE. The law now required that a hearing officer's ruling in a FAPE case "shall be made on *substantive grounds* [emphasis added] based on a determination of whether the child received a free appropriate public education" (IDEA, 20 U.S.C. 1415[f][3][E][i], 2004). Moreover, only certain procedural errors results in a substantive FAPE violation. To arise a substantive violation, the school districts errors must have: (I) impeded the child's right to a free appropriate public education; (II) significantly impeded the parents' opportunity to participate in the decision-making process regarding the provision of a free appropriate public education to the parents' child; or (III) caused a deprivation of educational benefits (IDEA, 20 U.S.C. 1415[f][3][E][ii], 2004). Additional guidance has been provided in U.S. District Court's *Kirby v. Cabell County Board of Education*, (D.WV. 2006). Despite this guidance, schools commonly make similar substantive errors, which are listed in Figure 2.

As Berney and Gilsbach (2017) pointed out, the statutory distinction made by Congress in 2004 codified the procedural and substantive distinction previously identified by the courts. Although the *Rowley/Endrew F.* test is primarily a tool to be used by IHOs, ALJs, and judges when determining FAPE, school administrators and teachers may also use the two-part test to assess the development and implementation of their students' IEPs. When using these tests, school personnel should ensure that the (a) procedural requirements of the IDEA and applicable state laws are followed and (b) the content of a student's IEP will enable him or her to make progress in light of the student's unique individual circumstances. The degree of progress will likely be made by examining the likely or actual results of a student's IEP (Hott et al. 2021; Zirkel 2019).

Common Substantive Errors

- Failing to conduct a comprehensive, individualized and relevant assessment.

- Failing to thoroughly evaluate the current performance of the child to determine PLAAFP statements.

- Failing to write ambitious, measurable annual goals.

- Failing to address all needs in the PLAAFP.

- Failing to link assessment, goals, and programming.

- Failing to provide relevant and meaningful special education and related services

- Failing to collect data to monitor student progress and make instructional changes when needed

- Failing to implement the IEP as written.

**Figure 2.** Substantive Errors.

As the original defining case of special education law that pertains to a measurement of FAPE, the procedural standard, or part one of the two-part test set by *Rowley* still applies with respect to ensuring that schools meet the procedural requirements of the IDEA (U.S. Department of Education 2017). The second part of the Rowley test, whether a student's IEP was reasonably calculated to enable the student to receive educational benefit, has been replaced by the *Endrew F.* standard that a student's IEP must be reasonably calculated to enable him or her to make progress appropriate in light of the student's circumstances. To meet this higher standard, school personnel must ensure that objectives and goals are ambitious enough to enable progress. Add to these procedural and substantive standards to which special educators must adhere, school district personnel must also implement student' IEP as agreed upon. We next provide guidance on how school administrators and teachers can develop and implement IEPs that meet the procedural, substantive, and implementation requirements of the IDEA.

**5. Developing IEPs That Meet the Procedural, Substantive, and Implementation Requirements of the IDEA**

The rulings we have reviewed have critical implications for special education administrators and teachers. We next point out the essential guidance provided in this decision.

**1. Ensure that school district officials and personnel understand and adhere to the procedural requirements of the IDEA.** Violation of certain procedural requirements, in and of themselves, could results in the denial of FAPE to a student with disabilities eligible for services under the IDEA. Foremost among these procedural violations that may violate a student's right to a FAPE is any misstep that effectively results in the student's parents not being involved in the special education decision-making process. A number of special education-related cases, including cases out of the U.S. Supreme Court, have confirmed the obligation of schools to ensure that parents participate in the special education process under the IDEA (Conroy and Yell 2019). The Ninth Circuit stressing the importance of involving parents from assessment to IEP formulation to implementation. In other words, a student's parents must be involved in the assessment, developing their child's IEP, and be provided with frequent and reports of their child's progress toward their annual goals. In an earlier case, the ninth circuit court also noted that parental rights are the "very essence of the IDEA" and if these rights are abridged in any way, it is very likely that the student

will be denied a FAPE. (*Amanda J. v. Clark County School District*, 2001, p. 892). Figure 3 is a checklist that school districts may follow to prevent such errors.

---

**Suggestions for a checklist to assist in avoiding procedural errors**

❑ Provide thorough training in procedural requirements to all involved staff.

❑ Focus on the basics (PWNs, consent, parent rights).

❑ Develop and use an agenda to guide the IEP meeting.

❑ Involve a students' parents in a meaningful way in decision-making meetings.

❑ Determine the placement after developing a student's IEP.

❑ Do not engage in predetermination of services or placement.

❑ Systematically take good notes at an IEP meeting.

❑ Identify a note-taker, not the LEA rep or SPED teacher.

---

**Figure 3.** Checklist, Procedural Errors.

**2. Ensure that special educators are skilled in developing IEPs that are reasonable calculated to enable students to make progress appropriate in light of their circumstances.** Students' IEPs must be written and implemented so as to enable them to meet the new, higher standard of educational benefit, which is the U.S. Supreme Court's *Endrew F.* test. According to Zirkel (2017) the substantive aspect of FAPE focus on the actual or likely results of student's IEPs. To ensure that IEPs meet this new standard, special educator administrators and teachers must ensure that their IEPs appropriately address the following requirements. First, IEPs must be based on full, individualized assessment of a student's academic and functional needs so that an IEP can be developed. Second, a student's present levels of academic achievement and functional performance (PLAAFP) statements, upon which a student's IEP are based, must, in effect, become a baseline by which a student's progress toward his or her goals can be measured and reported. Third, annual goals must be written so they are measurable. In general, a measurable goal should include the following components: (a) target behavior, (b) condition under which measurement will occur, and (c) criteria for acceptable performance (Yell 2019). Unfortunately, as Barbara Bateman observed in 2017, "too few IEP members know how to write measurable goals and too few goal writers intend that anyone ever actually measure the progress the child has made" (Bateman 2017, p. 98). A problem, which Bateman referred as "the cycle of non-accountability, (Bateman 2017, p. 98)" if not addressed often will lead to IEPs that will not meet the *Endrew F.* standard. If school district officials do not if not address the non-accountability cycle often will lead to IEPs that will not meet the *Endrew F.* standard. School district officials can best address these problems through training of administrators and teachers in their (a) obligations under the IDEA, and (b) methods to ensure that IEPs include relevant and thorough assessments and measurable and measured goals (Hott et al. 2021).

According to both the *Rowley* and the *Endrew F.*, rulings, the advancement from grades to grades are suitable progress for some students with disabilities but not for all students. In the U.S. Supreme Court 's opinion, Chief Justice of the John Roberts expressed this issue as follows:

Rowley sheds light on what appropriate progress will look like in many cases: For a child *fully integrated in the regular classroom* [emphasis added], an IEP typically should

be "reasonably calculated to enable the child to achieve passing marks and advance from grade to grade . . . . Rowley did not provide concrete guidance with respect to a child who is not fully integrated in the regular classroom and not able to achieve on grade level. *A child's IEP need not aim for grade-level advancement if that is not a reasonable prospect* [emphasis added]. But that child's educational program must be appropriately ambitious in light of his circumstances, just as advancement from grade to grade is appropriately ambitious for most children in the regular classroom. *The goals may differ, but every child should have the chance to meet challenging objectives* [emphasis added] (*Endrew. F.*, 2017, p. 922).

According to the *Endrew F.* FAPE standard a goal of advancing from grade to grade may not be a viable option; clearly some students with disabilities should have IEP goals that specifically address their unique academic and functional characteristics. Moreover, these students will require goals that are ambitious and challenging in accordance with their individualized needs. In such situations school officials could be required to "provide a cogent and responsive explanation for their decisions that shows the IEP is reasonably calculated to enable the child to make progress appropriate in light of his circumstances" (*Endrew F.*, 2017, p. 993). We suggest, therefore, that IEP teams craft measurable annual goals for students with disabilities that are ambitious, but reasonable.

Another lesson to be learned from the *Endrew F.* case is that IEP teams need to continually, at least annually, review the goals the have team has set for students. Even if a student has not yet completed a goal, IEP teams should continue to adjust goals so that they are ambitious and challenging. Refurbishing an IEP with the same or similar goals is often not enough to show ambitious growth. This does not mean old goals should be immediately disregarded but should be continually evaluated just as a student's progress toward a goal is monitored. For example, a goal that involved counting change could be adjusted to be more ambitious and focus on new, but similar, skills such as having the student identify the coins and ordering them in value.

As there is no way to predict every circumstance that will arise or the multitude of different goals, IEP teams may want to focus on some guiding questions such as*: Is the student's goal(s) ambitious enough for them?* If the goal is calculated to enable students to show progress throughout the year then team members need to ask themselves, *what data do we have that shows the student is in fact making or not making progress on this goal*? Finally, if a student has an ambitious goal and data, the final question to ask may be, *what is limiting progress for this student, and what can the district do to help alleviate that setback?* With these guiding ideas in mind, IEP teams may be able to avoid a situation similar to the one faced by the Douglas County School District in the *Endrew F.* case. Figure 4 is a checklist that school districts may follow to prevent such errors.

*Suggestions for a checklist to assist in avoiding substantive errors*

❑ Provide thorough training in substantive requirements to all involved staff

❑ Perform meaningful and relevant assessments of a student's unique needs

❑ Write individualized PLAAFP statements and connect to measurable goals and services

❑ Monitor students' progress and make changes when needed

❑ Implement the IEP as written.

**Figure 4.** Checklist, Substantive Errors.

## 6. Summary

*Rowley* (1982) and *Endrew F.* (2017) have very much shaped the special landscape with respect to the provision of FAPE to students with disabilities eligible under the IDEA. Although there is no bright line as to what a FAPE is for an individual student, hearing offers, administrative law judges, and state and federal judges are guided by the two-part *Rowley/Endrew F.* test in determining FAPE: (a) did the school district adhere to the procedural requirements of the IDEA, and (b) were students' IEPs reasonably calculated to enable them to make progress appropriate in light of their circumstances. Understanding and following this two-part test will help to ensure that school districts offer procedurally and substantively correct IEPs.

**Author Contributions:** Writing—original draft, M.R.; Writing—review & editing, M.L.Y. and J.W. All authors have read and agreed to the published version of the manuscript.

**Funding:** This research received no external funding.

**Institutional Review Board Statement:** Not applicable.

**Informed Consent Statement:** Not applicable.

**Data Availability Statement:** Not applicable.

**Conflicts of Interest:** The authors declare no conflict of interest.

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
