# Peer review of "Free Appropriate Public Education, the U.S. Supreme Court, and Developing and Implementing Individualized Education Programs"

_laws_

Round 1
Reviewer 1 Report
This is an outstanding piece of scholarship. It deals with a very significant issue in most educational systems, and it is extremely well-written, cogent, and persuasive. It should be published with high priority, as is.
Author Response
Per Reviewer #1, no changes suggested or required. Thank you.
Reviewer 2 Report
It is always a privilege to read work in special education law. See the PDF for feedback.

Round 2
Reviewer 2 Report
Addressed most concerns including identifying cases to support the procedural and substantive error figures. Would have preferred a more substantive review. However, it is well written and a general discussion of pitfalls that could benefit school leaders and other education professionals.
Author Response
We have addressed all five "minor edits" in the attached document. Please see the attachment. Thanks!
